# Phylogenetic diversity and virulence gene characteristics of *Escherichia coli* from pork and patients with urinary tract infections in Thailand

Pramualchai Ketkhao[1], Fuangfa Utrarachkij[1], Nattaya Parikumsil[2], Kritchai Poonchareon[3], Anusak Kerdsin[4], Peeraya Ekchariyawat[1], Pawarut Narongpun[5], Chie Nakajima[5,6], Yasuhiko Suzuki[5,6,7]*, Orasa Suthienkul[8]*

1 Faculty of Public Health, Department of Microbiology, Mahidol University, Bangkok, Thailand, 2 Photaram Hospital, Ratchaburi, Thailand, 3 Division of Biochemistry, School of Medical Sciences, University of Phayao, Phayao, Thailand, 4 Faculty of Public Health, Kasetsart University Chalermprakiat Sakon Nakhon Province Campus, Sakhon Nakhon, Thailand, 5 Division of Bioresources, Hokkaido University International Institute for Zoonosis Control, Sapporo, Japan, 6 International Collaboration Unit, Hokkaido University International Institute for Zoonosis Control, Sapporo, Japan, 7 Hokkaido University Institute for Vaccine Research and Development, Sapporo, Japan, 8 Faculty of Public Health, Mahidol University, Bangkok, Thailand

* suzuki@czc.hokudai.ac.jp (YS); orasa.sut@mahidol.ac.th (OS)

**Data Availability Statement:** All relevant data are within the manuscript.

## Abstract

Extraintestinal pathogenic *Escherichia coli* (ExPEC), especially uropathogenic *E. coli* (UPEC) are responsible for urinary tract infections (UTIs), while diarrheagenic *E. coli* (DEC) cause foodborne illnesses. These pathogenic *E. coli* are a serious threat to human health and a public concern worldwide. However, the evidence on pork *E. coli* (PEC) harboring UPEC virulence-associated genes is currently limited. Therefore, this study aimed to determine the phylogroups, virulence genes, and their association between PEC and UPEC from UTI patients. In this study, 330 *E. coli* were obtained from archived stock culture isolated from pork (PEC; n = 165) and urine of patients with UTIs (UPEC; n = 165) during 2014–2022. Phylogroups, UPEC- and diarrheagenic *E. coli* (DEC) associated virulence genes were assessed using PCR assays. The results showed that phylogroups A (50.3%), and B1 (32.1%) were commonly found among PEC whereas phylogroups B2 (41.8%), and C (25.5%) were commonly detected in the UPEC. PEC and UPEC carried similar virulence-associated genes with different percentages. The most frequent UPEC virulence-associated gene among UPEC, and PEC strains was *fimH*, (93.3%, and 92.1%), followed by *iucC* (55.2%, and 12.7%), *papC* (21.8%, and 4.2%), *afaC* (22.4%, and 0%), *hlyCA* (17%, and 0.6%), *cnf* (16.4%, and 0.6%), and *sfa/focDE* (8.5%, and 4.8%). Additionally, 6 of 27 UPEC virulence-associated gene patterns were found in both PEC and UPEC strains regardless of phylogroups. Furthermore, the DEC virulence-associated genes were found in only 3 strains, one from PEC harboring *eae*, and two from UPEC carried *fimH-bfpA* or *afaC-CVD432* indicating hybrid strains. Cluster analysis showed a relationship between PEC and UPEC strains and demonstrated that PEC harboring UPEC virulence-associated genes in pork may be associated with UPEC in humans. Food safety and hygiene practices during

**Funding:** This work was supported, in part, by a grant from the Ministry of Education, Culture, Sports, Science and Technology (MEXT), Japan, for the Joint Research Program of the Hokkaido University Research Center for Zoonosis Control toY.S., and, in part, by Japan Agency for Medical Research and Development (AMED) under Grant Numbers JP233fa627005 and JP20wm0125008 to Y.S.

**Competing interests:** The authors have declared that no competing interests exist.

pork production chain are important procedures for minimizing cross-contamination of these strains that could be transmitted to the consumers.

## Introduction

Pathogenic *Escherichia coli* are important Gram-negative bacteria that cause intestinal and extraintestinal diseases in humans and animals worldwide [1,2]. The pathogenicity of *E. coli* is determined by numerous virulence factors such as adhesion, invasion, colonization, and toxin secretion [3]. Intestinal *E. coli* are composed of non-pathogenic *E. coli* that are typically lacking virulence factors without causing gastroenteritis, and intestinal pathogenic *E. coli* (InPEC) or diarrheagenic *E. coli* (DEC) that carry various virulence factors causing a variety of gastrointestinal tract infections [4]. Extraintestinal pathogenic *E. coli* (ExPEC) carry virulence factors that commonly cause infections outside the intestinal tract and are mostly associated with urinary tract infections (UTIs). ExPEC can colonize the intestinal tract without causing gastroenteritis [5].

At present, InPEC or DEC are classified into eight pathotypes causing different types of gastroenteritis [4]. Enteropathogenic *E. coli* (EPEC) contain the intimin (*eae*) with or without bundle forming pili (*bfp*) for typical (tEPEC) or atypical (aEPEC) and are primarily associated with diarrhea in infants [6]. Enterotoxigenic *E. coli* (ETEC) cause illness through the production of heat-stable (*st*) and heat-labile (*lt*) enterotoxins causing traveler's diarrhea [7]. Shiga toxin-producing *E. coli* (STEC) produce Shiga toxins encoded by the *stx1* and *stx2* genes, while enterohaemorrhagic *E. coli* (EHEC) are defined by acquiring an additional intimin (*eae*) gene and a hemolysin (*hlyA*) gene. Both pathotypes are responsible for abdominal pain, bloody diarrhea, and are the primary causes of hemorrhagic colitis and hemolytic uremic syndrome (HUS) [8,9]. Enteroinvasive *E. coli* (EIEC) carry the *ipaH* gene, which is associated with an invasive plasmid antigen causing bloody diarrhea in infants [10]. Enteroaggregative *E. coli* (EAEC) contain the *aagR* gene, which encodes a transcriptional activator and is a major contributor to acute diarrhea in both children and adults globally [11]. Diffusely adherent *E. coli* (DAEC) possess the *afa* and *dra* genes, encoding afimbrial adhesins, and involved in sporadic cases of diarrhea, especially in pediatric patients [12]. Adherent invasive *E. coli* (AIEC) are associated with inflammatory bowel diseases, like Crohn's disease. However, AIEC are identified based on its phenotype due to the lack of a specific molecular markers [13]. Diarrheal diseases caused by DEC are a significant cause of morbidity and mortality in developing countries, especially in infants and young children [3]. ETEC, EPEC, and EHEC produce the same and different virulence factors in humans causing diarrheal diseases and also causing several diseases in industrious swine that resulted in the death of piglets [1]. Consequently, these pathotypes are possibly contaminated in animal-origin products including pork causing human diarrhea [4].

ExPEC are classified into four pathotypes namely uropathogenic *E. coli* (UPEC), newborn meningitis *E. coli* (NMEC), sepsis-associated *E. coli* (SEPEC), and avian pathogenic *E. coli* (APEC) [14]. UPEC are the majority of pathogens among ExPEC pathotypes. This pathotype can disseminate and colonize the urinary tracts of the hosts causing UTIs, affecting approximately 150 million cases globally each year [15,16]. Approximately 80% of UPEC infections cause community-acquired UTIs, but the sources of these UPEC remain unknown [17]. Two groups of essential virulence factors for UPEC to colonize and persist in the urinary tract are bacterial cell surface and secreted virulence factors. Virulence factors associated with bacterial cell surface include type 1 fimbriae (*fimH*), P fimbriae (*pap*), S fimbriae, and F1C fimbriae (*sfa/foc*). Hemolysin (*hlyCA*), and cytotoxic necrotizing factor 1 (*cnf1*) are secreted virulence

factors [15]. Moreover, hybrid pathogenic *E. coli* harboring both UPEC and DEC virulence-associated genes isolated from UTI patients have been reported, which may increase the potential severity of intestinal and extraintestinal infections [18].

Tracing the origin of ExPEC causing UTIs has still been discussed. Several studies have demonstrated that human UPEC have potentially originated from animals [19,20]. Some evidence demonstrated that UPEC virulence genes such as *fimH*, *pap*, *sfa/foc*, *ibeA* (invasin), and *iutA*, *iroN* (iron acquisition) from poultry and domestic companion animals were similar to human UPEC [19,21]. Recently, swine ExPEC causing UTIs in swine are increasingly found in the production and supply chain of pork in several countries [22–24]. Various virulence factors of the swine and pork ExPEC such as *fimH*, *iutA*, *sfa/foc*, and *kpsMTII* (capsular antigens) were similar to the human UPEC [22,24–27]. Furthermore, the same pulse-field gel electrophoresis (PFGE) patterns of *E. coli* strains from human UTI patients and pork have been reported, suggesting an increased risk of zoonotic infections [27]. These findings showed that swine and their products; pork may serve as a transmission source of UPEC and potentially cause UTIs in humans resulting in public health concerns. However, the information on the genetic virulence determinants related to swine and pork, human UPEC, and hybrid UPEC in Thailand was limited and has to be further elucidated.

Phylogroup identification by Clermont *et al.* [28] is essential and useful for epidemiological studies of *E. coli*. This method was applied to classify the various origins of *E. coli* into eight phylogroups: A, B1, B2, C, D, E, F, and cryptic clade I. The phylogroups have correlated with the presence of common virulence factors associated with ExPEC such as *papC*, *sfa/foc*, *cnf*, and *afa* (afimbriae adhesins). Most of the ExPEC/UPEC from humans were classified into phylogroups B2 and D while commensal *E. coli* belongs to phylogroups A and B1 [29,30]. However, there are very limited studies on the phylogroup classification of *E. coli* both from human UTIs, animals, and pork in Thailand.

Additional information regarding the contamination of pork with *E. coli* carrying UPEC virulence-associated genes, as well as hybrid UPEC strains, needs to be updated in Thailand. Therefore, this study aimed to elucidate the sharing status of UPEC and DEC virulence characteristics between pork *E. coli* (PEC) and UPEC from UTI patients. Additionally, the prevalence, distribution, virulence-associated genes, phylogroups, hybrid UPEC strains, and their association between PEC and UPEC from UTI patients were conducted to update the current situation in Thailand.

## Materials and methods

### Selection and isolation of *E. coli*

A total of 330 *E. coli* strains was obtained from the preserved stock culture collection of the Department of Microbiology, Faculty of Public Health, Mahidol University, during 2014–2022. All the strains in this study were retrieved from our stock cultures from two different sources: I) pork (n = 165) from local markets; and II) urine of UTI patients (n = 165) with counts number of *E. coli* $\geq 10^5$ CFU/ml from hospitals. The studied stock pork *E. coli* (PEC) and *E. coli* from UTI patients or UPEC were collected in many provinces. All archived *E. coli* strains were revived on MacConkey agar (Merck, Darmstadt, Germany), and biochemical confirmation using IMViC (indole, methyl red, Voges-Proskauer, and citrate) was described elsewhere [31].

### Genomic DNA extraction

All *E. coli* strains were extracted their genomic DNA by using the boiled-lysate method as previously described [32]. Briefly, 1 ml of overnight grown *E. coli* in Luria Bertani (LB) broth (BD

Biosciences, CA, USA) was centrifuged at 12,000 xg for 5 min. Then, the supernatant was discarded and 500 µl of TE buffer (10 mM Tris-HCl, 1 mM EDTA, pH 8) was added to re-suspend the pellet. The suspension was centrifuged at 12,000x g for 5 min and discarded the supernatant. One hundred microliter TE buffer was added to dissolve the pellet. The suspensions were boiled at 100˚C for 10 min then placed on ice for 10 min and centrifuged at 16,000x g for 5 min. The clear gDNA supernatant was obtained and kept as DNA templates at -20˚C for further analysis.

## Determination of phylogroups

Eight *E. coli* phylogroups including A, B1, B2, C, D, E, F, and clade I, were determined using quadruplex and singleplex PCR assays as previously described by Clermont *et al.* (2013). Genes for quadruplex PCR amplification included *chuA*, *yjaA*, *TspE4.C2*, and *arpA*. Primer sequences and PCR conditions were followed as previously described [28] (S1 Table). Both quadruplex and singleplex PCR mixtures contained 5 µl of 5X GoTaq reaction buffer (1.5 mM MgCl$_2$), 0.5 µl of dNTP mix (200 µM each), and 0.5 µl of 0.5 µM each forward and reverse primers, 0.125 µl of 0.125 units GoTaq DNA polymerase (Promega, WI, USA), 1 µl of the DNA template and adjusted to a final volume of 25 µl with deionized water.

Thermal cycling was performed in Thermo Hybaid Px2 Thermal Cycler (Thermo Fisher Scientific, MA, USA). Amplification conditions were as follows: initial denaturation at 95˚C for 2 min, 30 cycles of denaturation at 95˚C for 30 sec, annealing at the appropriate temperature of each primer pair for 30 sec, extension at 72˚C for 30 sec, and final extension at 72˚C for 5 min. Each specific temperature condition is described in S1 Table. The PCR products were analyzed by electrophoresis on 2% agarose gel in 1X TBE buffer and the amplicons were visualized under Blue Light LED Transilluminator (Dyne Bio, Seongnam, South Korea). The phylogroups were interpreted by the presence and absence of specific amplicons as shown in S2 Table.

## Detection of virulence-associated genes

Seven UPEC-associated virulence genes including *fimH*, *papC*, *sfa/focDE*, *afaC*, *hlyCA*, *cnf*, and *iucC*, and 10 DEC-associated virulence genes used for the detection of STEC (*stx1*, *stx2*), EPEC (*eae*, *bfpA*), ETEC (*lt*, *st*), EIEC (*ial*), EIEC (*ipaH*), EAEC (*CVD432*) and DAEC (*daaE*) were determined using single and multiplex PCR assays. The primer sequences of studied virulence genes and PCR conditions were as previously described [33–36] (S1 Table).

Each PCR mixture of both single and multiplex PCR contained 5 µl of 5X GoTaq reaction buffer (1.5 mM MgCl$_2$), 0.5 µl of dNTP mix (200 µM each), and 0.5 µl of 0.5 µM each forward and reverse primers, 0.125 µl of 0.125 units GoTaq DNA polymerase (Promega), 1 µl of the DNA template, and adjusted to a final volume of 25 µl with deionized water. Thermocycling was carried out in Thermo Hybaid Px2 Thermal Cycler (Thermo Fisher Scientific) as follows: 95˚C for 2min followed by 30 cycles of 95˚C for 30sec, the appropriate annealing temperature for each primer pair for 30s, and 72˚C for 1min followed by a final extension step of 72˚C for 5min (S1 Table). Amplicons were analyzed as described above.

## Cluster analysis of virulence genes and phylogroups

To perform cluster analysis, the presence or absence of UPEC and DEC virulence-associated genes and phylogroups were converted to a zero-one matrix. Then, the analysis was constructed with UPGMA (Unweighted Pair Group Method for Arithmetic Averages) clustering based on Dice's correlation coefficients with a default bootstrap value of 100 using the

DendroUPGMA [37]. A dendrogram of all strains was visualized using Interactive Tree Of Life (iTol) version 6.8.1 [38].

## Statistical analysis

The prevalence of phylogroups and virulence-associated gene distribution were expressed as numbers and percentages. The data were analyzed using Statistical Package for Social Sciences software version 26 (SPSS, IL, USA). The prevalence of phylogroups, virulence-associated genes, and their association between PEC and UPEC were compared using the Chi-square or Fisher's exact. The $p$-value $< 0.05$ was considered statistically significant.

# Results

## Distribution of phylogroups

The phylogroups were determined in 165 PEC strains and 165 UPEC strains. Among 165 PEC strains positive for phylogroup A (50.3%; 83/165) were found in the highest prevalence, followed by B1 (32.1%; 53/165). The low detection rates ranging from 0.6% to 5.5% were found for phylogroups C, D, F, clade I, and B2 (Fig 1). In contrast, the most common phylogroup among 165 UPEC was B2 (41.8%; 69/165), followed by C (25.5%; 42/165) (Fig 1). The clade I group was not found in all tested UPEC strains. When compared to the detection rates between PEC and UPEC strains in phylogroups, phylogroups A and B1 in PEC strains were significantly higher than those in UPEC ($p<0.05$) while phylogroups B2 and C in UPEC were significantly higher than those in PEC strains ($p<0.05$), and phylogroups D and F in both strains were not significantly different ($p>0.05$).

## Virulence-associated genes

Amplifying seven UPEC virulence-associated genes, including *fimH*, *papC*, *sfa/focDE*, *afaC*, *hlyCA*, *cnf*, and *iucC*, in 330 *E. coli* strains showed that all the genes were found in both PEC and UPEC strains, except for *afaC* gene that was not found in all PEC strains. The most frequent gene was *fimH* (92.7%; 306/330), followed by *iucC* (33.9%; 112/330) (Fig 2). The similar detection rates of *fimH* were found for PEC (92.1%; 152/165) and UPEC (93.3%; 154/165). The frequencies of PEC and UPEC strains positive for at least one UPEC virulence gene were 92.7% (153/165) and 98.2% (162/165), respectively. In contrast, significant differences between PEC and UPEC strains that harbored UPEC virulence-associated genes were found in this study ($p<0.05$). Moreover, the detection rates of PEC strains positive for the following virulence-associated genes were significantly lower than those in UPEC strains: *iucC* (12.7% < 55.2%), *papC* (4.2% < 21.8%), *hlyCA* (0.6% < 17%), and *cnf* (0.6% < 16.4%) ($p<0.05$) (Fig 2).

Most of the strains (99.1%; 327/330) were negative for all DEC virulence-associated genes, namely *stx1*, *stx2*, *bfpA*, *eae*, *lt*, *st*, *ial*, *ipaH*, *CVD432*, and *daaE*. Only one PEC strain harbored *eae* for aEPEC. Additionally, two UPEC strains were positive for *bfpA* (aEPEC), and *CVD432* (EAEC) genes. These two UPEC strains were considered hybrid strains. A UPEC strain carried *fimH-bfpA* gene and belonged to phylogroup B2 (*chuA*, *TspE4C2*) while another UPEC harbored *afaC* was positive for *CVD432* gene and belonged to phylogroup D (*arpA*, *chuA*, *TspE4C2*).

## Association between the phylogroups and UPEC virulence-associated genes

The association between the phylogroups and the seven UPEC virulence-associated genes was investigated in this study (Table 1). Among PEC strains, phylogroups were not significantly associated with the presence of most UPEC virulence-associated genes ($p>0.05$). However,

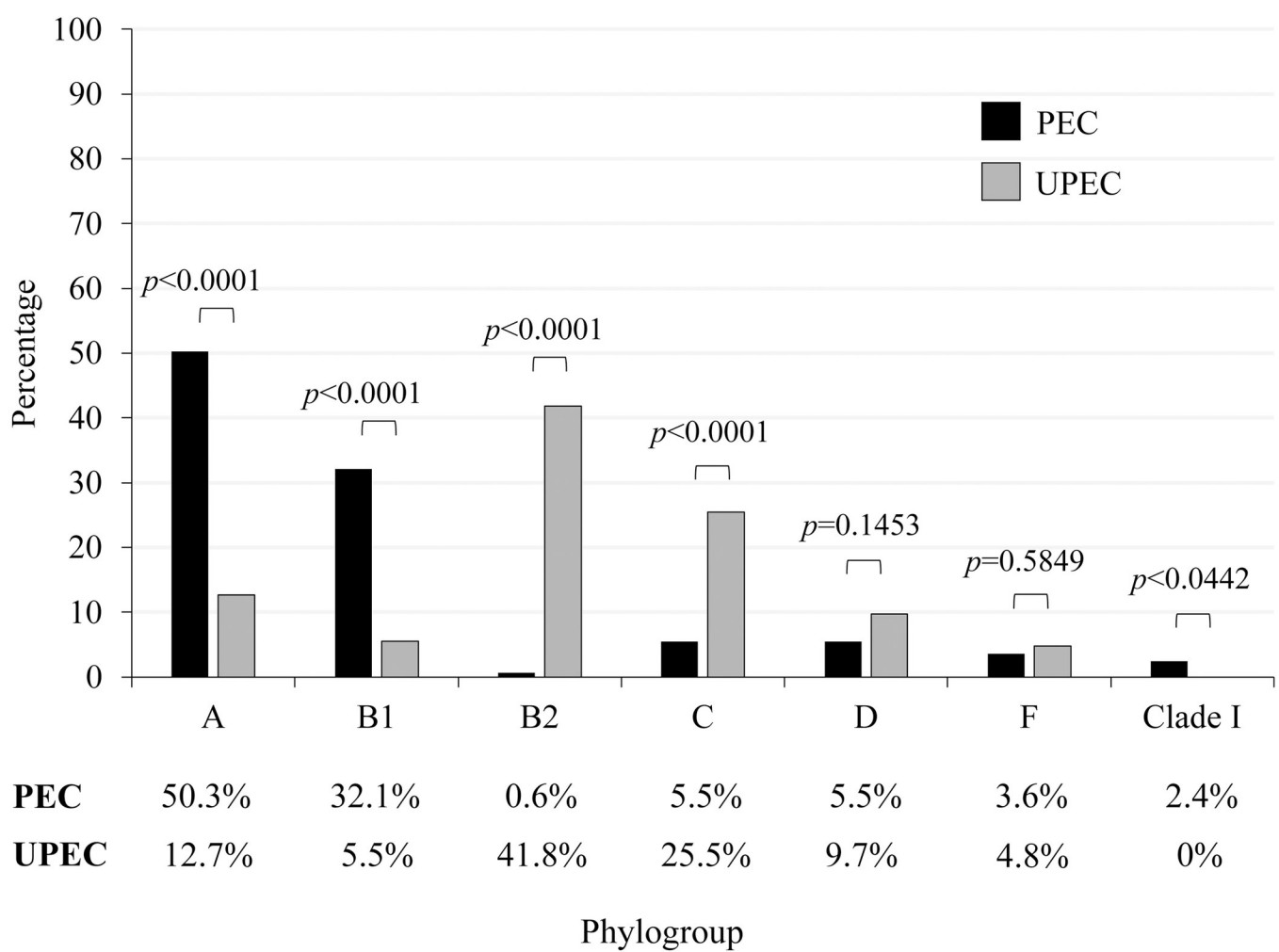

**Fig 1. Distribution of the phylogroups among *Escherichia coli* isolated from pork (PEC) and urine (UPEC) of patients with urinary tract infections (UTIs) in Thailand.**

*fimH* was found the association with phylogroups C and clade I ($p = 0.0035$, $p = 0.0015$, respectively) while *papC* was significantly associated with phylogroup F ($p = 0.0214$). Moreover, PEC strains belonging to phylogroups B1 and F were significantly associated with *iucC* ($p = 0.0044$, $p = 0.0243$, respectively) (Table 1). Among UPEC strains, the results showed that phylogroup B1 was significantly associated with *sfa/focDE* ($p = 0.0304$). Phylogroups B2 and C were significantly associated with *iucC*, *papC*, and *afaC* ($p<0.05$) (Table 1).

## UPEC virulence-associated gene patterns distribution among phylogroups of PEC and UPEC strains

Based on the distribution of various UPEC virulence-associated genes, the studied strains were divided into 27 virulence gene patterns (Table 2). Of 27 patterns, pattern No. 25 (*sfa/focDE*⁺ *cnf*⁺ *hlyCA*⁺ *fimH*⁺ *iucC*⁺), was only found among PEC strains (0.6%; 1/165) belonging to phylogroup B1, while 19 patterns were found only in UPEC strains (45.5%; 75/165). Six patterns were found both in PEC (92.1%; 152/165) and UPEC strains (52.7%; 87/165). The pattern No. 4 (*fimH*⁺) was the most commonly found in PEC strains (80%; 132/165) while the pattern No.

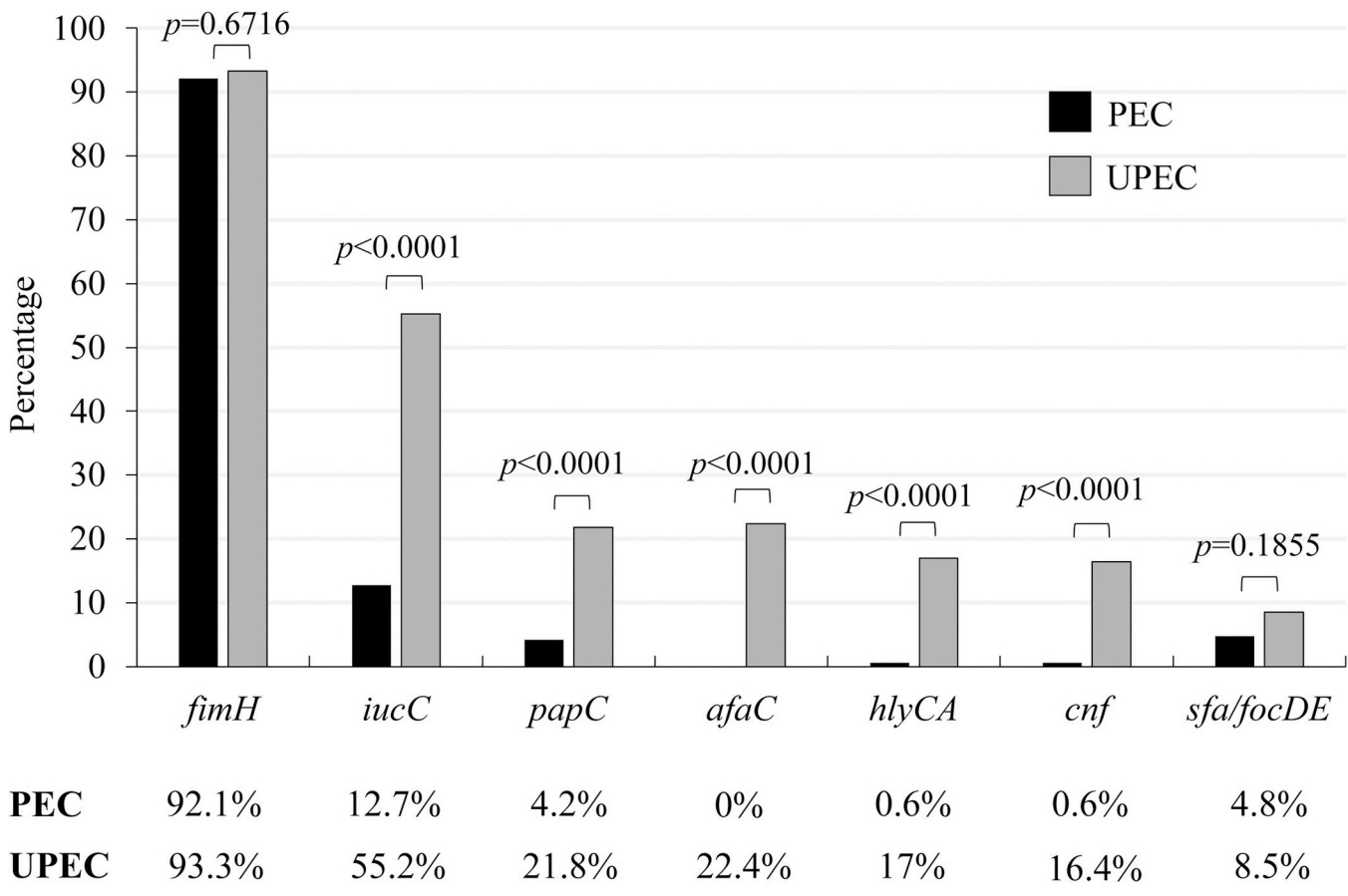

| | *fimH* | *iucC* | *papC* | *afaC* | *hlyCA* | *cnf* | *sfa/focDE* |
|---|---|---|---|---|---|---|---|
| **PEC** | 92.1% | 12.7% | 4.2% | 0% | 0.6% | 0.6% | 4.8% |
| **UPEC** | 93.3% | 55.2% | 21.8% | 22.4% | 17% | 16.4% | 8.5% |

UPEC virulence-associated gene

**Fig 2. The distribution of uropathogenic *Escherichia coli* (UPEC) virulence-associated genes of *E. coli* from pork (PEC) and urine (UPEC) of patients with urinary tract infections (UTIs) in Thailand.**

12 (*fimH*[+] *iucC*[+]) was the major pattern found in UPEC strains (25.5%; 42/165). Moreover, the pattern No. 4 (*fimH*[+]) was found in all phylogroups (80%; 132/165) except for B2 among PEC strains while phylogroup A was the most frequent phylogroup (47.3%; 78/165) among PEC strains. The pattern No. 12 (*fimH*[+] *iucC*[+]), the most commonly found among UPEC strains, was also found in all phylogroups except for clade I and B2 (15.2%; 25/165) (Table 2).

## Cluster analysis of virulence genes and phylogroups

The UPGMA cluster analysis was used to categorize individual strains into larger clusters and generate a dendrogram illustrating the relationships between PEC and UPEC strains. Based on the presence or absence of DEC and UPEC virulence genes in each strain and phylogroups, eight main clusters and several sub-clusters were found in this study (Fig 3). The most predominant groups were cluster III (31.2%; 103/330) and cluster VIII (21.2%; 70/330). Among the cluster III, 80.6% (83/103) were PEC while 19.4% (20/103) were UPEC. All strains in this cluster belonged to phylogroup A and had fewer UPEC virulence-associated genes. A single UPEC virulence gene, *fimH*, was commonly found in this cluster. Moreover, an aEPEC strain from PEC (*eae*) was found in this cluster. In contrast, most of the strains in the cluster VIII

**Table 1. Frequencies of virulence factors associated with uropathogenic *Escherichia coli* (UPEC) in the phylogroups among *Escherichia coli* isolated from pork (PEC) and urine (UPEC) of patients with urinary tract infections (UTIs) in Thailand.**

| Virulence gene | No. (%) of strain positive for phylogroup | | | | | | | | | | | | | |
|---|---|---|---|---|---|---|---|---|---|---|---|---|---|---|
| | A | | B1 | | B2 | | C | | D | | F | | Clade I | |
| | PEC (n = 83) | UPEC (n = 21) | PEC (n = 53) | UPEC (n = 9) | PEC (n = 1) | UPEC (n = 69) | PEC (n = 9) | UPEC (n = 42) | PEC (n = 9) | UPEC (n = 16) | PEC (n = 6) | UPEC (n = 8) | PEC (n = 4) | UPEC (n = 0) |
| *fimH* | 78 (94) | 17 (81) | 52 (98.1) | 9 (100) | 1 (100) | **69 (100)**[a] | **5 (55.6)**[a] | 42 (100) | 9 (100) | 13 (81.3) | 6 (100) | 4 (50) | **1 (25)**[a] | **0** |
| *iucC* | 0 | 9 (42.9) | **12 (22.6)**[a] | 3 (33.3) | 1 (100) | **58 (84.1)**[a] | 3 (33.3) | **7 (16.7)**[a] | 2 (22.2) | 7 (43.8) | **3 (50)**[a] | 7 (87.5) | 0 | **0** |
| *papC* | 0 | 2 (9.5) | 3 (5.7) | 1 (11.1) | 1 (100) | **29 (42)**[a] | 1 (11.1) | **1 (2.4)**[a] | 0 | 2 (12.5) | **2 (33.3)**[a] | 1 (12.5) | 0 | **0** |
| *afaC* | 0 | 3 (14.3) | 0 | 0 | 0 | **5 (7.2)**[a] | 0 | **24 (57.1)**[a] | 0 | 4 (25) | 0 | 1 (12.5) | 0 | **0** |
| *hlyCA* | 0 | 0 | 1 (1.9) | 0 | 0 | **28 (40.6)**[a] | 0 | 0 | 0 | 0 | 0 | 0 | 0 | **0** |
| *cnf* | 0 | 0 | 1 (1.9) | 0 | 0 | **27 (39.1)**[a] | 0 | 0 | 0 | 0 | 0 | 0 | 0 | **0** |
| *sfa/focDE* | 0 | 1 (4.8) | 5 (9.4) | **3 (33.3)**[a] | 1 (100) | 9 (13) | 1 (11.1) | 1 (2.4) | 0 | 0 | 1 (16.7) | 0 | 0 | **0** |

[a]significantly associated with UPEC virulence-associated genes ($p<0.05$).

were from UPEC (98.6%; 69/70) which belonged to phylogroup B2 and carried multiple UPEC virulence-associated genes. However, only one PEC (1.4%; 1/70) was found. Quintuple and sextuple UPEC virulence-associated genes were frequently found in cluster VIII. A hybrid strain (*fimH-bfpA*) from UPEC was also found in cluster VIII (Fig 3).

## Discussion

To our knowledge, this is the first study on the determination of *E. coli* phylogroups and virulence genes between PEC and UPEC in Thailand. Phylogroup analysis data in this study showed that phylogroups A (50.3%) and B1 (32.1%) were commonly found among PEC while phylogroups B2 (41.8%) and C (25.5%) were the most prevalent among UPEC (Fig 1). It is known that *E. coli* from phylogroups A and B1 are often commensal [28], while human UPEC strains causing UTIs belong to phylogroups B2 and D [39,40]. Therefore, most of the PEC and UPEC in this study were commensal and ExPEC, respectively. However, the frequency of phylogroup C (25.5%) among UPEC strains in the present study was higher than phylogroup D (9.7%). This result differed from previous Thai studies of human UPEC which belonged to phylogroups B2 and D [41,42]. From the phylogenetic tree analysis of several *E. coli*, phylogroup C was close to phylogroup B1 [28]. However, a low frequency of phylogroup B1 (5.5%) was found in our study indicating that phylogroup C may dominate among UPEC in Thailand instead of phylogroup D. Moreover, phylogroups F and clade I were found in PEC. Phylogroup F is related to phylogroup B2 [28] while clade I is frequently found in human and animal feces [43]. The presence of these new phylogroups might resulted from using the new quadruplex PCR assay [28] which could be a more accurate assignment than the former assay [44].

All seven UPEC virulence-associated genes could be found in UPEC strains in our study (Fig 2). More than half (55.2%) of UPEC strains in the present study carried *iucC* essential for the survival of UPEC inside urinary tracts under an iron-limiting environment [15]. Several studies from different countries including Thailand were also reported high prevalences of this

**Table 2. Uropathogenic *Escherichia coli* (UPEC) virulence gene patterns among phylogroups of 330 *Escherichia coli* from pork (PEC) and urine (UPEC) of patients with urinary tract infections (UTIs) in Thailand.**

| Pattern No. | Virulence gene | | | | | | | Phylogroup (No. of strain (PEC/UPEC)) | | | | | | | No. of strain (%) | | |
|---|---|---|---|---|---|---|---|---|---|---|---|---|---|---|---|---|---|
| | *afaC* | *sfa/focDE* | *papC* | *cnf* | *hlyCA* | *fimH* | *iucC* | A | B1 | B2 | C | D | F | Clade I | PEC n = 165 | UPEC n = 165 | Total N = 330 |
| 1 | | | | | | | | 5/1 | 1/- | | 3/- | -/2 | | 3/- | 12 (7.3) | 3 (1.8) | 15 (4.5) |
| 2 | ■ | | | | | | | 1 | | | | 1 | | | 0 | 2 (1.2) | 2 (0.6) |
| 3 | | ■ | | | | | | 1 | | | | | | | 0 | 1 (0.6) | 1 (0.3) |
| 4 | | | | | | ■ | | 78/8 | 40/4 | -/3 | 3/11 | 7/5 | 3/1 | 1/- | 132 (80) | 32 (19.4) | 164 (49.7) |
| 5 | | | | | | | ■ | | | | | | 3 | | 0 | 3 (1.8) | 3 (0.9) |
| 6 | ■ | | | | ■ | | | | | | 22 | 1 | | | 0 | 23 (13.9) | 23 (7) |
| 7 | ■ | | | | | ■ | | | | | | | 1 | | 0 | 1 (0.6) | 1 (0.3) |
| 8 | | ■ | | | | ■ | | | 1 | | 1 | | | | 0 | 2 (1.2) | 2 (0.6) |
| 9 | | | ■ | | | ■ | | 1 | 1 | | 1 | | | | 0 | 3 (1.8) | 3 (0.9) |
| 10 | | | ■ | | | | ■ | -/1 | | | 1/- | | | | 1 (0.6) | 1 (0.6) | 2 (0.6) |
| 11 | | | | | ■ | ■ | | | | 1 | | | | | 0 | 1 (0.6) | 1 (0.3) |
| 12 | | | | | | ■ | ■ | -/6 | 5/1 | -/25 | 1/5 | 2/3 | 1/2 | | 9 (5.5) | 42 (25.5) | 51 (15.5) |
| 13 | ■ | | | | | ■ | ■ | 2 | 4 | 2 | 2 | | | | 0 | 10 (6.1) | 10 (3) |
| 14 | | ■ | ■ | | | ■ | | | 1 | | | | | | 0 | 1 (0.6) | 1 (0.3) |
| 15 | | ■ | | | | ■ | ■ | 3/2 | | 1/- | | | | | 4 (2.4) | 2 (1.2) | 6 (1.8) |
| 16 | | | ■ | | | ■ | | 2/- | | -/5 | | -/2 | 1/1 | | 3 (1.8) | 8 (4.8) | 11 (3.3) |
| 17 | | | | ■ | | | | | | | 1 | | | | 0 | 1 (0.6) | 1 (0.3) |
| 18 | | ■ | ■ | | | ■ | | 1/- | 1/2 | | | | 1/- | | 3 (1.8) | 2 (1.2) | 5 (1.5) |
| 19 | | ■ | | ■ | ■ | ■ | | | 1 | | | | | | 0 | 1 (0.6) | 1 (0.3) |
| 20 | | | ■ | ■ | ■ | ■ | | | 1 | | | | | | 0 | 1 (0.6) | 1 (0.3) |
| 21 | | | ■ | | ■ | ■ | | | 1 | | | | | | 0 | 1 (0.6) | 1 (0.3) |
| 22 | | | | ■ | ■ | ■ | | | | | 4 | | | | 0 | 4 (2.4) | 4 (1.2) |
| 23 | ■ | | | ■ | ■ | ■ | | | 1 | | | | | | 0 | 1 (0.6) | 1 (0.3) |
| 24 | | ■ | ■ | ■ | ■ | | | | | -/4 | | | | | 0 | 4 (2.4) | 4 (1.2) |
| 25 | | ■ | | ■ | ■ | ■ | ■ | 1 | | | | | | | 1 (0.6) | 0 | 1 (0.3) |
| 26 | | | ■ | ■ | ■ | ■ | | | | 14 | | | | | 0 | 14 (8.5) | 14 (4.2) |
| 27 | | ■ | ■ | ■ | ■ | ■ | | | | 1 | | | | | 0 | 1 (0.6) | 1 (0.3) |

gene [29,41,45]. Much higher frequencies of *hlyCA* (17.0%) and *cnf* (16.4%) were found among UPEC than PEC in this study. However, those from UPEC observed in this study were higher than in the previous study from Thailand (7% and 1%, respectively) [40]. These genes are involved in tissue damage and have been linked to patients with pyelonephritis rather than cystitis [46,47]. Although low frequencies of these genes were found among PEC, this strain may have the ability to cause UTIs. Regarding fimbrial adhesins among UPEC, the prevalences of *afaC* and *papC* in our study were 22.4% and 21.8%, respectively. These frequencies were lower than the previous report (79%) in Thai UPEC [48]. The *papC* and *afaC* are more associated with upper urinary tract infections due to their ability to promote UPEC attach more specifically to kidneys and renal tubules [49]. Therefore, the discrepancy between our results and previous reports from Thailand may be isolated from the patients with different types of UTIs.

 All the UPEC virulence-associated genes were found among PEC, except for *afaC* gene (Fig 2). It is worth noting that these PEC strains shared several UPEC virulence-associated genes with human strains [34]. In contrast, high frequencies of UPEC virulence genes namely *papC* (58%), *cnf* (23.2%), and *afa* (13.4%) were found in the urine of diseased sows in China [23]. Slaughtering and dressing are critical steps for cross-contamination of these strains from urine

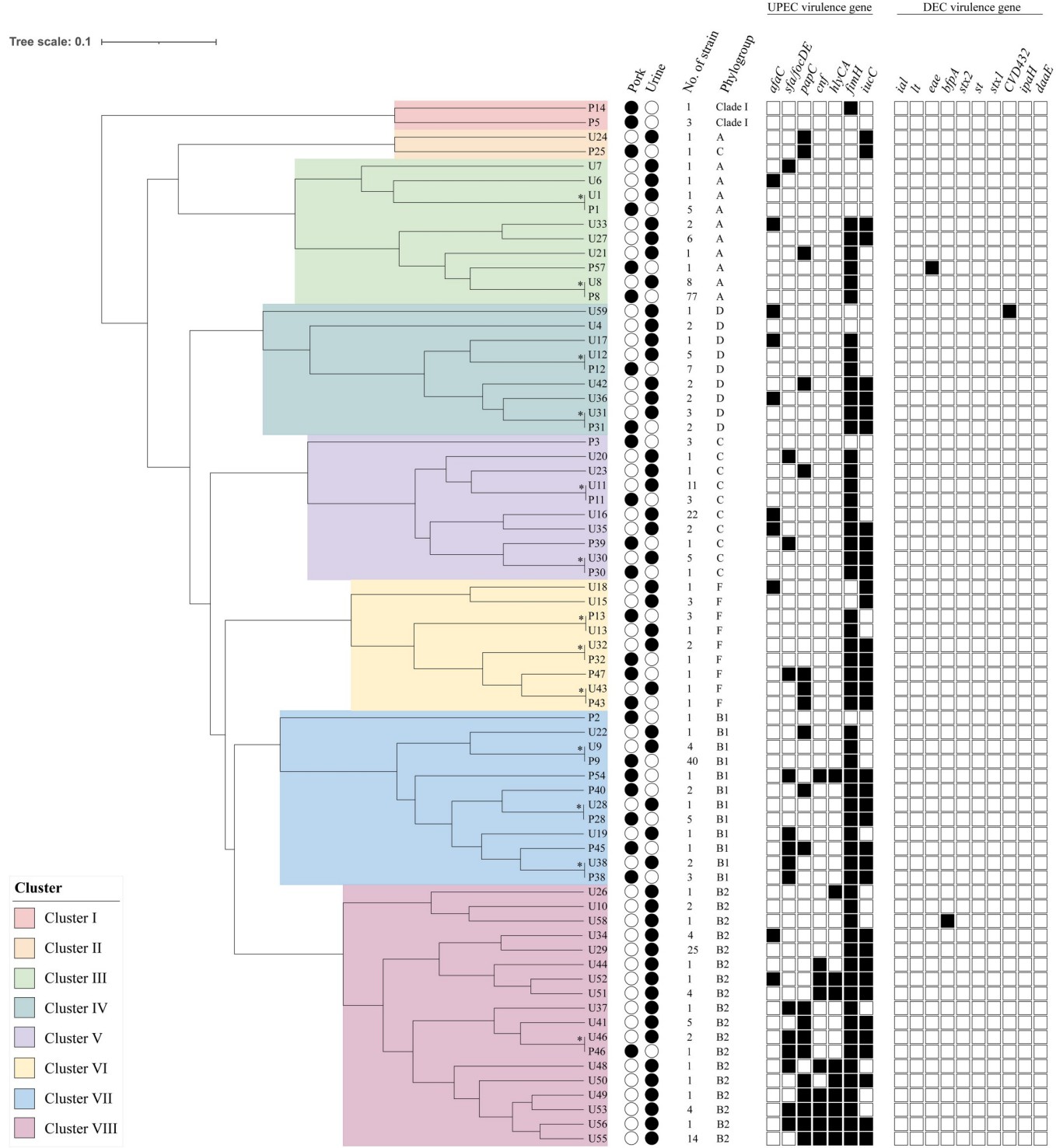

**Fig 3. UPGMA clustering analysis of 330 *Escherichia coli* from pork (PEC) and urine (UPEC) of patients with urinary tract infections (UTIs) in Thailand.** The dendrogram was constructed based on phylogroups and presence/absence of uropathogenic *E. coli* (UPEC) and diarrheagenic *E. coli* (DEC) virulence-associated genes. Asterisks indicate PEC and UPEC were found in the same lineage. Black or white circles represent strains from pork or urine, respectively. Groups of the same profile strains are collapsed and shown proportionately to the number of strains in that lineage. White or black boxes indicate the absence or presence of virulence genes, respectively. No cut-off value was defined; clustering was based on total differences found.

to pork that may serve as potential sources of the UPEC strains. Several PEC strains shared similar UPEC virulence-associated gene patterns to UPEC strains in this study (Table 2). Moreover, multiple UPEC virulence-associated genes were found among these PEC strains (highlighted in grey in Table 2). The UPEC strains obtained from patients with pyelonephritis had a greater number of virulence genes than those isolated from patients with cystitis [46]. Various UPEC virulence-associated genes such as *fimH*, *sfa*, *foc*, *afa*, *hly*, and *cnf* are located on pathogenicity islands and could be transferred to other strains through horizontal transfer [50]. Moreover, healthy human intestines colonized by several UPEC strains have been reported [51]. Therefore, gut commensal strains might have gained UPEC virulence-associated genes from PEC with those, making them a possible cause of UTIs. Therefore, consuming undercooked and contaminated pork may increase the risk of colonization of these PEC carrying UPEC virulence genes followed by accidental infections in urinary systems [52].

Only three strains in the present study carried the DEC virulence-associated genes and two of these strains were considered as hybrid strains due to the presence of both UPEC and DEC virulence-associated genes. The UPEC in this study harbored *fimH-bfpA* (UPEC/aEPEC) while the other UPEC carried *afaC-CVD432* (UPEC/EAEC). The UPEC/EAEC was more frequently found among UTI patients with diarrhea and associated with a higher risk of developing pyelonephritis than those without diarrhea [53]. These strains may have acquired UPEC properties and become a potential cause of either UTIs, diarrhea in humans, or both. Females have a higher risk of developing these infections due to the short distance between the rectum and the urethra [54]. However, this finding needs to be further investigated due to limited information on the hybrid strains in Thailand.

To investigate the possibility of an association between phylogroups and UPEC virulence-associated genes, *E. coli* strains from phylogroup B2 had the highest frequency of virulence-associated genes (Table 1). This result is in good agreement with previous studies in Thailand [48] and Mexico [45]. However, several UPEC virulence-associated genes were found in the PEC belonging to phylogroups A and B1 in this study suggesting that phylogroups A and B1 regarded as commensal strains might be the reservoirs of the UPEC virulence-associated genes. Moreover, these strains could be opportunistic pathogens for the development of UTIs in humans.

The UPGMA clustering results showed that the PEC or UPEC strains in this study were grouped into eight main clusters (Fig 3). Some of the PEC and UPEC strains were found in the same clusters and some strains showed similar genotypes of UPEC virulence-associated genes and phylogroups. For example, 77 PEC and 8 UPEC belonging to phylogroup A and carrying *fimH* were found in the same lineage (with asterisks indicated in Fig 3). This finding may imply that several UPEC strains in this study were closely related to PEC as previously reported in some human UPEC and animal ExPEC [17,18,52]. The PEC harbored UPEC virulence-associated genes may pose a zoonotic risk and may have a relationship to UPEC in humans. These PEC strains may have potentially caused UTIs in humans after colonizing the outside of gastrointestinal tracts. To confirm this hypothesis, a more comprehensive investigation into the molecular epidemiology of these strains is necessary. This would involve analyzing genetic similarities and evolutionary relationships using both basic and advanced molecular techniques.

## Conclusions

To our knowledge, this is a novel study that provides information on phylogroups, UPEC and DEC virulence-associated genes, and the association between *E. coli* from UTI patients and pork in Thailand. Several UPEC virulence-associated genes were found in PEC strains and

shared similar sets of virulence genes to human UPEC strains. Our study highlights that PEC carrying UPEC virulence-associated genes may serve as reservoirs of virulence-associated genes and further transmit to humans causing UTIs. Additional studies are needed to further determine the relationship of genetic clones by the molecular epidemiology among these strains. However, this finding is raising more attention to great concern for food safety, and public health perspectives.

## Supporting information

**S1 Table. PCR primers used in this study.**
(PDF)

**S2 Table. Interpretation of amplified PCR products for phylogroups of Escherichia coli.**
(PDF)

## Acknowledgments

We would like to acknowledge the great contributions of the late Prof. Lee Riley, Chair of the Infectious Disease and Vaccinology Division, UC Berkeley School of Public Health, for his valuable and constructive suggestions during the planning and development of this research.

## Author Contributions

**Conceptualization:** Fuangfa Utrarachkij, Yasuhiko Suzuki, Orasa Suthienkul.

**Data curation:** Pramualchai Ketkhao.

**Formal analysis:** Pramualchai Ketkhao.

**Funding acquisition:** Yasuhiko Suzuki.

**Investigation:** Pramualchai Ketkhao, Fuangfa Utrarachkij, Orasa Suthienkul.

**Methodology:** Pramualchai Ketkhao, Fuangfa Utrarachkij, Orasa Suthienkul.

**Resources:** Pramualchai Ketkhao, Nattaya Parikumsil, Kritchai Poonchareon, Anusak Kerdsin, Orasa Suthienkul.

**Supervision:** Fuangfa Utrarachkij, Chie Nakajima, Yasuhiko Suzuki, Orasa Suthienkul.

**Validation:** Pramualchai Ketkhao, Fuangfa Utrarachkij.

**Visualization:** Pramualchai Ketkhao.

**Writing – original draft:** Pramualchai Ketkhao.

**Writing – review & editing:** Fuangfa Utrarachkij, Nattaya Parikumsil, Kritchai Poonchareon, Anusak Kerdsin, Peeraya Ekchariyawat, Pawarut Narongpun, Chie Nakajima, Yasuhiko Suzuki, Orasa Suthienkul.

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
