## [Decision Letter · Decision Letter 0]

22 Mar 2024

PONE-D-23-44006Phylogenetic diversity and virulence gene characteristics of Escherichia coli from pork and patients with urinary tract infections in ThailandPLOS ONE

Dear Dr. Suzuki,

Thank you for submitting your manuscript to PLOS ONE. After careful consideration, we feel that it has merit but does not fully meet PLOS ONE’s publication criteria as it currently stands. Therefore, we invite you to submit a revised version of the manuscript that addresses the points raised during the review process.

We look forward to receiving your revised manuscript.

Kind regards,

Kwame Kumi Asare, Ph.D

Academic Editor

PLOS ONE

“This work was supported, in part, by a grant from the Ministry of Education, Culture, Sports, Science and Technology (MEXT), Japan, for the Joint Research Program of the Hokkaido University Research Center for Zoonosis Control toY.S., and, in part, by Japan Agency for Medical Research and Development (AMED) under Grant Numbers JP20jk0210005, JP20jm0110021, and JP20wm0125008 to Y.S.”

“We would like to acknowledge the great contributions of the late Prof. Lee Riley, Chair of the Infectious Disease and Vaccinology Division, UC Berkeley School of Public Health, for his valuable and constructive suggestions during the planning and development of this research. Consumable reagents were partly supported by the Department of Microbiology, Faculty of Public Health, Mahidol University. This work was supported, in part, by a grant from the Ministry of Education, Culture, Sports, Science and Technology (MEXT), Japan, for the Joint Research Program of the Hokkaido University Research Center for Zoonosis Control toY.S., and, in part, by Japan Agency for Medical Research and Development (AMED) under Grant Numbers JP20jk0210005, JP20jm0110021, and JP20wm0125008 to Y.S.”

“This work was supported, in part, by a grant from the Ministry of Education, Culture, Sports, Science and Technology (MEXT), Japan, for the Joint Research Program of the Hokkaido University Research Center for Zoonosis Control toY.S., and, in part, by Japan Agency for Medical Research and Development (AMED) under Grant Numbers JP20jk0210005, JP20jm0110021, and JP20wm0125008 to Y.S.”

Reviewers' comments:

Reviewer's Responses to Questions

**Comments to the Author**

1. Is the manuscript technically sound, and do the data support the conclusions?

Reviewer #1: Yes

Reviewer #2: Yes

Reviewer #3: Partly

Reviewer #4: Partly

Reviewer #5: No

Reviewer #6: Partly

Reviewer #7: Yes

Reviewer #8: No

2. Has the statistical analysis been performed appropriately and rigorously? 

Reviewer #1: Yes

Reviewer #2: Yes

Reviewer #3: Yes

Reviewer #4: Yes

Reviewer #5: Yes

Reviewer #6: Yes

Reviewer #7: Yes

Reviewer #8: Yes

3. Have the authors made all data underlying the findings in their manuscript fully available?

Reviewer #1: Yes

Reviewer #2: Yes

Reviewer #3: Yes

Reviewer #4: Yes

Reviewer #5: Yes

Reviewer #6: Yes

Reviewer #7: Yes

Reviewer #8: Yes

4. Is the manuscript presented in an intelligible fashion and written in standard English?

Reviewer #1: Yes

Reviewer #2: Yes

Reviewer #3: Yes

Reviewer #4: Yes

Reviewer #5: Yes

Reviewer #6: Yes

Reviewer #7: Yes

Reviewer #8: Yes

5. Review Comments to the Author

Reviewer #1: -The introduction should provide more context and background information on the significance of studying the transmission of virulence genes between pork and urinary pathogenic E. coli strains.

-In materials and methods, Were there any specific criteria used for selecting pork samples and UTI patient samples for inclusion in the study?

-The utilization of UPGMA clustering for analyzing the presence of virulence-associated genes and phylogroups is appropriate. However, the manuscript lacks clarity regarding the rationale for selecting this specific clustering method. Were alternative methods considered, and if so, why were they deemed less suitable? Additionally, the authors should provide more details regarding the interpretation of the dendrogram generated from the clustering analysis to aid readers in understanding the relationships between strains.

-We suggest that you refer to new research article doi: 10.3389/fcimb.2022.790184. and doi: 10.1007/s11033-023-09031-x in the discussion section.

Reviewer #2: Dear Authors,

I would like to say this study is very neat and has informative data that will be of interest for international readers of the journal.

Before publication, I recommend following my comment below:

Line 60: This paragraph provide some information about different E. coli pathotypes, I would suggest to use more references and not only one. I suggest to use this study: 10.30466/vrf.2022.550618.3416 as the second references for this paragraph.

Reviewer #3: Comments

*What were the aim and conclusion of your study, exactly? It is not clear to me the aim of the study on sharing virulence genes between PEC and UPEC. Did you aim to demonstrate that pork may serve as one possible source of UPEC? If so, how did you demonstrate that in the present work?

[Cluster analysis … demonstrated that pork contaminated with E. coli harboring UPEC virulence genes may serve as one possible source of UPEC in humans],[This finding may imply…. UPEC strains in this study were closely related to PEC]. [Our study highlights that PEC carrying UPEC virulence-associated genes may………further transmit to humans causing UTIs].

* Genotyping of isolates is needed to obtain accurate data and true prevalence. It is possible that your strains originated from a clone.

*[A total of 330 E. coli strains was obtained from the preserved stock culture collection]:

did you have exactly the same number of PEC and UPEC in your collection or you include 165 isolates (each group) from this collection in your study?

*What was your definition for UPEC?

*Grammatical errors and writing issues were seen in the manuscript.

*For abbreviations, provide their full term for the first mention.

*Information on the isolates is not provided. How were urines collected? What was the demographic data of patients?

* FimH is not only for UPEC. It is found in other E. coli types and in E. coli belonging to normal flora.

*Which tests were used to identify E. coli? Mention them.

*Did you test the antibiotic susceptibility of isolates?

*Some references such as reference 1 were referred to in the manuscript too much. Use other references instead of referring to 1.

*Figure titles need to be revised.

*Remove Table 1 and embed its data within the text

*Table 2: number of isolates is not enough to make a p-value, in some rows.

Best

Reviewer #4: Major concerns:

Please include the criteria used to characterize strains as UPEC, PEC and DEC in the Methods. A description of the approach is given in the Introduction (Lines 55 and 110), but the specific genomic markers used to make the determination should be listed. This impacts the finding on lines 184-185 that “165 isolates PEC and 165 ones UPEC” The collection criteria stated on lines 117-124 only select for an isolation source and make no distinction between UPEC, E.coli commensal or environmental contaminant. If the authors were able to subset the UPEC strains from this dataset, please provide the selection criteria. Otherwise, it would be more accurate to claim that the 330 isolates clustered with at least eight different groups of isolates of UPEC and PEC source. Further, were any tests performed to determine if isolates E1-165 were the causative agents of infection? If not, then additional justification is needed to categorize them as UPEC.

Reviewer #5: Firstly, I would like to thank you for the opportunity and trust in my work to review this scientific article. Below, I provide some comments:

Line 124:

I believe it would be interesting to describe the biochemical tests for E. coli characterization. Was there also molecular confirmation of the species?

The dendrogram image is of low quality, and it was not possible to analyze it.

Although there is a phylogenetic analysis in the paper (represented by the dendrogram that could not be analyzed), I believe that epidemiological data from the samples obtained would be necessary for the inferences made.

The work is interesting, but it should focus on the actual evidence obtained. The samples were obtained from a previous isolate bank. For the conclusions reached, it would be necessary to analyze epidemiological data that were not presented.

I suggest that epidemiological data be presented or that the conclusions and discussions obtained be reviewed.

I also suggest that a higher quality image of the dendrogram be provided.

Reviewer #6: Comment to the authors

Abstract

Have only the specific genes of UPEC been examined? The text raises this suspicion. If not, please change the text of the first lines in the abstract.

Line 36 – 40: The percentage of the genes in the UPEC is higher than in PEC, so it is better to present its name and its percentage first.

The conclusion of the abstract section is not based on the results and should be changed

Introduction

Line 56-59: First introduce EXPEC more. Mention and explain the virulence genes or pathogenic factors of this bacterium that cause urinary infections.

Line 67: Please write in full version for “STEC” for the first time.

Line 76: State the pathotypes of EXPEC here with more details along with its pathogenic genes.

Line 107-108: Please remove this sentence as a novelty, the authors have mentioned it several times in the text.

Material and method

More data should be provided on the sources of isolation of these bacteria, especially on human samples.

Does the study have a code of ethics, mention it.

The bootstrap value should be stated at the level of phylogenetic analysis.

How many times the same branch is observed when repeating the generation of a phylogenetic tree on a resampled set of data? The percentage of times each interior branch is given a value of 1 should be noted.

Reviewer #7: The paper entitled “Phylogenetic diversity and virulence gene characteristics of Escherichia coli from pork and patients with urinary tract infections in Thailand” has good and important subject, good writing, and interesting results. However some comments are as follow:

The phylogenetic method designed by Clermont et al has different protocol. Why did the authors change the PCR protocol?

In material and method, the authors described that the cluster analysis was performed according to phtlogenetic group and absence or presence of virulence genes, while in the fig 3. the strains with similarity profiles put in different clusters. I think the clustering is drawn according to phylogenetic groups only.

Page 23 Line 293. What is the reason for this difference?

Page 24 Lines 318 -319 are repeated.

Page 24 Lines 347-348 “Females have a higher risk of developing these infections due to the short distance between the rectum and the urethra” What is the connection between this sentence and the previous sentences?

Reviewer #8: The present manuscript evaluated the diversity of phylogenetic groups and virulence genes of E. coli isolates from pork (PEC) and human patients with urinary tract infection (UPEC) in Thailand. The objectives of the work were to compare the virulence genotype of UPEC and PEC. The authors reported that the PEC strains were grouped predominantly in groups A and B1 (commensal strains), while the UPEC strains, which carry a greater number of virulence genes, were classified in phylogroups B2 and C. In conclusion, they pointed to pork as a reservoir of genes associated with UTI, reporting a major public health concern.

The paper has some important limitations that prevent it from being accepted in this format. The main ones are:

1- There is no coherence between the results and the conclusions. How can we say that this is a major public health risk if the profile of the strains in the two groups was different? How can the risk be measured? It would be more appropriate to use more modern analysis tools such as sequencing types or dendrograms generated by comparing WGS data. The frequency of genes between the two groups is important information, but on its own it doesn't serve to support the hypothesis or measure the risk.

2- The Clermont classification used is outdated. Currently, 9 phylogroups are considered. See Clermont et al. (2019) - Group G.

3- There is no hybrid of DAEC and UPEC, as the fimH gene is not a UPEC marker. On the contrary, it is a highly prevalent fimbria in enterobacteria that can be detected in non-pathogenic strains of the microbiota. This information needs to be reviewed and is considered a lack of definition of the concept of hybrid strains.

4- The statistics between the frequency of virulence genes and phylogroups is completely unnecessary information, given that the concept of phylogroups is based on the evolutionary process of acquiring virulence genes in the B2 group. So this result is to be expected and has no significance whatsoever in light of the proposal.

5- The results of group C are of local epidemiological data, bute the importance should be understood in more in-depth studies in Thailand.

So, I think this paper needs to be modified prior the publication.

6. PLOS authors have the option to publish the peer review history of their article (what does this mean?). If published, this will include your full peer review and any attached files.

Reviewer #1: **Yes: **Mehrdad Halaji

Reviewer #2: **Yes: **Mahdi Askari Badouei

Reviewer #3: No

Reviewer #4: **Yes: **Hami Kaboosi

Reviewer #5: No

Reviewer #6: No

Reviewer #7: No

Reviewer #8: **Yes: **Terezinha Knöbl

---

## [Author Response · Author response to Decision Letter 0]

14 Apr 2024

Responses to Reviewer #1

Thank you for giving us your valuable comments. I am sure our manuscript will be much improved by these.

Comment 1: The introduction should provide more context and background information on the significance of studying the transmission of virulence genes between pork and urinary pathogenic E. coli strains.

Response 1: More background information was added to the introduction part. Please go through it.

Comment 2: In materials and methods, Were there any specific criteria used for selecting pork samples and UTI patient samples for inclusion in the study?

Response 2: The E. coli strains were randomly selected from four regions in Thailand. Both pork and urine strains were selected in the same amount in each region.

Comment 3: The utilization of UPGMA clustering for analyzing the presence of virulence-associated genes and phylogroups is appropriate. However, the manuscript lacks clarity regarding the rationale for selecting this specific clustering method. Were alternative methods considered, and if so, why were they deemed less suitable? Additionally, the authors should provide more details regarding the interpretation of the dendrogram generated from the clustering analysis to aid readers in understanding the relationships between strains.

Response 3: The dendrogram was constructed by zero-one matrix data (binary data) from the phylogroups and virulence associated genes profiles using the unweighted pair group method with arithmetic averages (UPGMA) method via an online tool (www.genoms.urv.cat/upgma). Alternative methods such as sequence-based methods have been considered, but this study did not perform it due to the limited data. Therefore, the UPGMA was suitable for the available data in this study. More details of cluster analysis were added to materials and methods at Lines 181-186.

Comment 4: We suggest that you refer to new research article doi: 10.3389/fcimb.2022.790184. and doi: 10.1007/s11033-023-09031-x in the discussion section.

Response: The research article “10.3389/fcimb.2022.790184.” already mentioned in the discussion. Moreover, the article “10.1007/s11033-023-09031-x” was added to this section accordingly.

Responses to Reviewer #2: 

Thank you for giving us your valuable comment. I am sure our manuscript will be much improved by this.

Comment 1: Line 60: This paragraph provide some information about different E. coli pathotypes, I would suggest to use more references and not only one. I suggest to use this study: 10.30466/vrf.2022.550618.3416 as the second references for this paragraph.

Response 1: More information about different E. coli pathotypes was added into the introduction section (Line 60-82). Moreover, the suggested article was used as a reference.

Responses to Reviewer #3: Comments

Thank you for giving us your valuable comments. I am sure our manuscript will be much improved by these.

Comment 1: What were the aim and conclusion of your study, exactly? It is not clear to me the aim of the study on sharing virulence genes between PEC and UPEC. Did you aim to demonstrate that pork may serve as one possible source of UPEC? If so, how did you demonstrate that in the present work?

[Cluster analysis … demonstrated that pork contaminated with E. coli harboring UPEC virulence genes may serve as one possible source of UPEC in humans],[This finding may imply…. UPEC strains in this study were closely related to PEC]. [Our study highlights that PEC carrying UPEC virulence-associated genes may………further transmit to humans causing UTIs].

Response 1: Yes, this study aimed to demonstrate that pork may serve as one possible source of UPEC in humans. The presence of virulence genes in pork strains in this study may imply that these strains may serve as a possible source of UPEC in humans. Moreover, the results from cluster analysis showed that some strains of pork and urine samples were clustered in the same branch of the tree. However, a molecular epidemiology study of these strains will be confirmed in further study.

Comment 2: Genotyping of isolates is needed to obtain accurate data and true prevalence. It is possible that your strains originated from a clone.

Response 2: Our 330 E. coli strains were obtained from different pure stock cultures. A single colony of each E. coli strain was selected for the analysis. These strains must have originated from different clones. However, the similar profiles of pork strains and urine strains found in this study may or may not be the same clone. Therefore, the whole genome sequencing of these strains needed to be performed in further study.

Comment 3: [A total of 330 E. coli strains was obtained from the preserved stock culture collection]: did you have exactly the same number of PEC and UPEC in your collection or you include 165 isolates (each group) from this collection in your study?

Response 3: The 330 E. coli strains were selected from different 330 samples which included 165 pork samples and 165 urine samples.

Comment 4: What was your definition for UPEC?

Response 4: The definition of UPEC in this study was E. coli that isolated from urine of UTI patients with the count number of the E. coli ≥ 105 CFU/ml.

Response 5: Grammatical errors and writing issues were seen in the manuscript.

Response 5: Some grammatical errors and writing issues have been corrected.

Comment 6: For abbreviations, provide their full term for the first mention.

Response 6: The abbreviations have been corrected.

Comment 7: Information on the isolates is not provided. How were urines collected? What was the demographic data of patients?

Response 7: The demographic data of patients could not be provided because all urine strains were selected from stock cultures and human ethics were not available for this study. 

Comment 8: FimH is not only for UPEC. It is found in other E. coli types and in E. coli belonging to normal flora.

Response 8: Our results agree with that fimH is not only for UPEC. However, several studies included this gene for UPEC virulence-associated gene determination. It is one of the essential genes for attaching to epithelial cells and has been closely associated with the early development of UTIs (10.1073/pnas.0902179106, 10.1111/j.1758-2229.2010.00166.x, 10.1086/315258). Therefore, fimH was used to determine UPEC virulence-associated genes in this study. Moreover, E. coli originally isolated from the urine of patients with UTIs with counts number of E. coli ≥ 105 CFU/ml were considered UPEC strains in this study.

Comment 9: Which tests were used to identify E. coli? Mention them.

Response 9: The identification of E. coli was determined using morphological of colonies on MacConkey agar and confirmed by biochemical tests (IMViC; indole, methyl red, Voges-Proskauer, and citrate). Moreover, the PCR method was double-confirmed at the step of virulence gene determination using the PCR amplification of the 16S rRNA gene of E. coli (S1 Table). The detail of E. coli identification was added to the manuscript at Lines 135-136.

Comment 10: Did you test the antibiotic susceptibility of isolates?

Response 10: The antibiotic susceptibility of isolates will be performed in further study. This study focused on the phylogenetic diversity and virulence gene profiles of two sources of isolates and needed to emphasize the relationship between the strains from pork and urine.

Comment 11: Some references such as reference 1 were referred to in the manuscript too much. Use other references instead of referring to 1.

Response 11: The other references were used instead of using only reference 1.

Comment 12: Figure titles need to be revised.

Response 12: Fig 1 title was changed to “Distribution of the phylogroups among Escherichia coli isolated from pork (PEC) and urine (UPEC) of patients with urinary tract infections (UTIs) in Thailand.” and Fig 3 title was changed to “Fig 3. UPGMA clustering analysis of 330 Escherichia coli from pork (PEC) and urine (UPEC) of patients with urinary tract infections (UTIs) in Thailand.”.

Comment 13: Remove Table 1 and embed its data within the text

Response 13: Table 1 has been removed and the data was embedded into the text at Lines 232-234.

Comment 14: Table 2: number of isolates is not enough to make a p-value, in some rows.

Response 14: The data in Table 2 has been corrected.

Responses to Reviewer #4 

Thank you for giving us your valuable comments. I am sure our manuscript will be much improved by these.

Comment 1: Please include the criteria used to characterize strains as UPEC, PEC and DEC in the Methods. 

Response 1: In this study, UPEC were the E. coli originally isolated from the urine of patients with UTIs with counts number of E. coli ≥ 105 CFU/ml. For PEC, these E. coli were isolated pork meat. The DEC strains were identified using specific DEC virulence-associated genes. The details were added into the Selection and isolation of E. coli section (Line 132-133).

Comment 2: A description of the approach is given in the Introduction (Lines 55 and 110), but the specific genomic markers used to make the determination should be listed. 

Response 2: The specific genomic markers used to make the determination were listed into introduction (Lines 61-74 for DEC virulence-associated genes and Lines 90-93 for UPEC virulence-associated genes).

Comment 3: This impacts the finding on lines 184-185 that “165 isolates PEC and 165 ones UPEC” The collection criteria stated on lines 117-124 only select for an isolation source and make no distinction between UPEC, E.coli commensal or environmental contaminant. If the authors were able to subset the UPEC strains from this dataset, please provide the selection criteria. Otherwise, it would be more accurate to claim that the 330 isolates clustered with at least eight different groups of isolates of UPEC and PEC sources. Further, were any tests performed to determine if isolates E1-165 were the causative agents of infection? If not, then additional justification is needed to categorize them as UPEC.

Response 3: UPEC were the E. coli originally isolated from the urine of patients with UTIs with counts number of E. coli ≥ 105 CFU/ml while PEC were the E. coli isolated pork meat obtained from markets (The more information was added into the Selection and isolation of E. coli section at Lines 130-133). Eight clusters in this study were constructed based on phylogroups and virulence genes. Therefore, the sources of isolates were not used for this analysis. The UPEC Pork strains that exhibited similar phylogroups and virulence gene profiles to urine strains in UTI patients could not be confirmed as the causative agents of UTIs in this study. However, the genetic relationships between these strains will be performed using WGS in further study.

Responses to Reviewer #5

Thank you for giving us your valuable comments. I am sure our manuscript will be much improved by these.

Comment 1: Line 124: I believe it would be interesting to describe the biochemical tests for E. coli characterization. Was there also molecular confirmation of the species?

Response 1: Details of the biochemical test were added to the materials and methods (Lines 135-136). Moreover, the molecular confirmation of E. coli was also used at step virulence-associated gene determination by PCR amplification 16S rRNA of E. coli (S1 Table).

Comment 2: The dendrogram image is of low quality, and it was not possible to analyze it.

Although there is a phylogenetic analysis in the paper (represented by the dendrogram that could not be analyzed), I believe that epidemiological data from the samples obtained would be necessary for the inferences made.

Response 2: The new high quality dendrogram image was re-submitted. The epidemiological data from samples was not available for this study because all strains were selected from stock cultures.

Comment 3: The work is interesting, but it should focus on the actual evidence obtained. The samples were obtained from a previous isolate bank. For the conclusions reached, it would be necessary to analyze epidemiological data that were not presented. I suggest that epidemiological data be presented or that the conclusions and discussions obtained be reviewed. I also suggest that a higher quality image of the dendrogram be provided.

Response 3: The revised manuscript included the reviewer's suggestion.

Responses to Reviewer #6 

Thank you for giving us your valuable comments. I am sure our manuscript will be much improved by these.

Abstract

Comment 1: Have only the specific genes of UPEC been examined? The text raises this suspicion. If not, please change the text of the first lines in the abstract.

Response 1: Several UPEC virulence-associated genes have been examined, such as csgA, csgE, fimA, fimH, papA, papC, papGIII, kpsmII, iutA, cnf, hlyD, sfaS, focG, and afa (https://doi.org/10.1111/tbed.13848;
https://doi.org/10.1016/j.jmii.2016.03.006). However, there have been few studies examining these virulence genes among E. coli from pork samples.

Comment 2: Line 36 – 40: The percentage of the genes in the UPEC is higher than in PEC, so it is better to present its name and its percentage first.

Response 2: The revised manuscript included the reviewer's suggestion (Lines 37-40).

Comment 3: The conclusion of the abstract section is not based on the results and should be changed

Response 3: “These strains may pose health risks for the development of UTIs in humans” (I think this sentence should be removed according to the comment). The revised manuscript included the reviewer's suggestion (Lines 45-46).

Introduction

Comment 4: Line 56-59: First introduce EXPEC more. Mention and explain the virulence genes or pathogenic factors of this bacterium that cause urinary infections.

Response 4: The virulence genes and virulence factors causing UTIs in humans were described at Lines 83-85 and 90-93.

Comment 5: Line 67: Please write in full version for “STEC” for the first time.

Response 5: The full version of Shiga toxin-producing E. coli (STEC) was added in to the manuscript at Line 65.

Comment 6: Line 76: State the pathotypes of EXPEC here with more details along with its pathogenic genes.

Response 6: The revised manuscript included the reviewer's suggestion (Lines 83-85).

Comment 7: Line 107-108: Please remove this sentence as a novelty, the authors have mentioned it several times in the text.

Response 7: The revised manuscript included the reviewer's suggestion (Lines 118-119).

Material and method

Comment 8: More data should be provided on the sources of isolation of these bacteria, especially on human samples. Does the study have a code of ethics, mention it.

Response 8: The urine and pork strains in this study were selected from bacterial stock cultures. Therefore, human and animal ethics are not available for this study.

Comment 9: The bootstrap value should be stated at the level of phylogenetic analysis.

Response 9: The dendrogram was constructed using a bootstrap value of 100 based on the default settings of the program. The details of cluster analysis have been added into materials and methods (Lines 182-186).

Comment 10: How many times the same branch is observed when repeating the generation of a phylogenetic tree on a resampled set of data? The percentage of times each interior branch is given a value of 1 should be noted.

Response 10: The times of same branch were not available because this dendrogram is constructed using a bootstrap value of 100 based on the default settings of the program.

Responses to Reviewer #7

Thank you for giving us your valuable comments. I am sure our manuscript will be much improved by these.

Comment 1: The phylogenetic method designed by Clermont et al has different protocol. Why did the authors change the PCR protocol?

Response 1: We followed the original protocol for phylogrouping from Clermont et al. (2013) and did not change the PCR protocol. 

Comment 2: In material and method, the authors described that the cluster analysis was performed according to phtlogenetic group and absence or presence of virulence genes, while in the fig 3. the strains with similarity profiles put in different clusters. I think the clustering is drawn according to phylogenetic groups only.

Res

---

## [Decision Letter · Decision Letter 1]

10 May 2024

PONE-D-23-44006R1Phylogenetic diversity and virulence gene characteristics of Escherichia coli from pork and patients with urinary tract infections in ThailandPLOS ONE

Dear Dr. Suzuki,

Thank you for submitting your manuscript to PLOS ONE. After careful consideration, we feel that it has merit but does not fully meet PLOS ONE’s publication criteria as it currently stands. Therefore, we invite you to submit a revised version of the manuscript that addresses the points raised during the review process.

We look forward to receiving your revised manuscript.

Kind regards,

Kwame Kumi Asare, Ph.D

Academic Editor

PLOS ONE

Reviewers' comments:

Reviewer's Responses to Questions

**Comments to the Author**

1. If the authors have adequately addressed your comments raised in a previous round of review and you feel that this manuscript is now acceptable for publication, you may indicate that here to bypass the “Comments to the Author” section, enter your conflict of interest statement in the “Confidential to Editor” section, and submit your "Accept" recommendation.

Reviewer #1: All comments have been addressed

Reviewer #2: All comments have been addressed

Reviewer #3: (No Response)

Reviewer #4: All comments have been addressed

Reviewer #6: (No Response)

2. Is the manuscript technically sound, and do the data support the conclusions?

Reviewer #1: Yes

Reviewer #2: Yes

Reviewer #3: Partly

Reviewer #4: Yes

Reviewer #6: Partly

3. Has the statistical analysis been performed appropriately and rigorously? 

Reviewer #1: Yes

Reviewer #2: Yes

Reviewer #3: Yes

Reviewer #4: N/A

Reviewer #6: Yes

4. Have the authors made all data underlying the findings in their manuscript fully available?

Reviewer #1: Yes

Reviewer #2: Yes

Reviewer #3: Yes

Reviewer #4: Yes

Reviewer #6: No

5. Is the manuscript presented in an intelligible fashion and written in standard English?

Reviewer #1: Yes

Reviewer #2: Yes

Reviewer #3: Yes

Reviewer #4: Yes

Reviewer #6: No

6. Review Comments to the Author

**Reviewer #1:** This article is accepted in this form

This article is accepted in this form

This article is accepted in this form

**Reviewer #2:** The authors addressed all the questions properly, and I would like to recommend for acceptance of this article.

**Reviewer #3:** The authors made an extensive effort to improve the manuscript. There are still some ambiguities.

*Sharing virulence genes between E. coli groups is important data, but it can’t support your hypothesis. The dendrogram and cluster analysis of virulence genes also can’t be used to support the aim of the present study. Therefore, please change and revise the corresponding lines.

[Comment 1: What were the aim and conclusion of your study, exactly? ….…. Response 1 : Yes, this study aimed to demonstrate that pork may serve as one possible source of UPEC in humans. The presence of virulence genes in pork strains in this study may imply that these strains may serve as a possible source of UPEC in humans. Moreover, the results from cluster analysis showed that some strains of pork and urine samples were clustered in the same branch of the tree. However, a molecular epidemiology study of these strains will be confirmed in further study.]

*Some simpler and cheaper methods than WGS could be used to analyze the relationships between the isolates. Furthermore, the genotyping may help to support the hypothesis that :pork may serve as one possible source of UPEC in humans

*Please add more references to the first paragraph in Introduction: lines 51-59.[Comment 11: Some references were referred to in the manuscript too much. Use other references instead of referring to 1. Response 11: The other references were used instead of using only reference 1].

**Reviewer #4:** The authors have responded to the reviewer queries adequately and implemented the changes that have improved the accuracy and readability of the manuscript. I am satisfied with the changes implemented by the authors in this revised manuscript. However there are some more needs to revise the manuscript. First, Fig. 2 should be transferred to supplementary file. Second, in dendrogram it is unknown that what is the tolerance and optimization rate in analysis. There are some strains whose patterns are the same with 100 % similarity. But the dendrogram clustered them into different patterns. Like strains U20 and U19, also strains U21 and U22. Or some bands have been missed between isolates like strains U16 and U35 between P39 and P30 (U30). It seems that the authors should analyze the bands again with new tolerance and less sensitivity. So that after analysis these strains will got into the same pattern. After re-analysis the dendrogram all the text should be revised based on the results.

**Reviewer #6:** At first, the authors did not adequately answer some questions. For example, one of the questions was why the researchers used the method of UPGMA clustering for cluster analysis and why the other clustering methods were not suitable. They suggested that due to the limited facilities, it was not possible to perform other tests such as a complete genome examination, which was a wrong answer to this question. In fact, they should have explained other clustering methods, and why they did not use those methods. Another example is my question “Have only the specific genes of UPEC been examined?” they didn’t answer it correctly. They stated that the purpose of their study was to examine virulence genes related to UPEC isolates, but they have other types of E. coli isolates. In this section, writing should have been done in such a way that the importance of all isolate types was also shown.

7. PLOS authors have the option to publish the peer review history of their article (what does this mean?). If published, this will include your full peer review and any attached files.

Reviewer #1: No

Reviewer #2: **Yes: **Mahdi Askari Badouei

Reviewer #3: **Yes: **Safoura Derakhshan

Reviewer #4: **Yes: **Hami Kaboosi

Reviewer #6: No

---

## [Author Response · Author response to Decision Letter 1]

22 May 2024

Reviewer #1: 

Comment: This article is accepted in this form

Response: Thank you for your acceptance. We appreciate much.

Reviewer #2: 

Comment: The authors addressed all the questions properly, and I would like to recommend for acceptance of this article.

Response: Thank you for your acceptance. We appreciate much.

Reviewer #3: 

Comment 1: What were the aim and conclusion of your study, exactly? ….…. Response 1 : Yes, this study aimed to demonstrate that pork may serve as one possible source of UPEC in humans. The presence of virulence genes in pork strains in this study may imply that these strains may serve as a possible source of UPEC in humans. Moreover, the results from cluster analysis showed that some strains of pork and urine samples were clustered in the same branch of the tree. However, a molecular epidemiology study of these strains will be confirmed in further study.

Response: The conclusion of this study in the abstract section was revised to “Cluster analysis showed a relationship between PEC and UPEC strains and demonstrated that PEC harboring UPEC virulence-associated genes in pork may be associated with UPEC in humans” (Line 44-46). Moreover, the conclusion in the discussion part of cluster analysis was revised to “The PEC harbored UPEC virulence-associated genes may pose a zoonotic risk and may have a relationship to UPEC in humans.” (Lines 370-371).

Comment 2: Some simpler and cheaper methods than WGS could be used to analyze the relationships between the isolates. Furthermore, the genotyping may help to support the hypothesis that :pork may serve as one possible source of UPEC in humans

Response: The suggested methods for analyzing the relationship between the isolates were revised (Lines 373-375).

Comment 3: Please add more references to the first paragraph in Introduction: lines 51-59.[Comment 11: Some references were referred to in the manuscript too much. Use other references instead of referring to 1. Response 11: The other references were used instead of using only reference 1].

Response: References were accordingly in the Introduction (Line 51-59)

Reviewer #4: 

Comment: Fig. 2 should be transferred to supplementary file. Second, in dendrogram it is unknown that what is the tolerance and optimization rate in analysis. There are some strains whose patterns are the same with 100 % similarity. But the dendrogram clustered them into different patterns. Like strains U20 and U19, also strains U21 and U22. Or some bands have been missed between isolates like strains U16 and U35 between P39 and P30 (U30). It seems that the authors should analyze the bands again with new tolerance and less sensitivity. So that after analysis these strains will got into the same pattern. After re-analysis the dendrogram all the text should be revised based on the results.

Response: Thank you for your insightful suggestions. Our responses are as below.

1. Regarding your suggestion to transfer Fig. 2 to the supplementary file, we understand your suggestion. However, we would like to demonstrate the importance of Fig. 2 in illustrating the distribution of UPEC virulence-associated genes among pork and urine strains in Thailand. Given its significance for readers' understanding of the main study's findings, we believe it would be beneficial to keep Fig. 2 in the main manuscript. I hope you will consider this rationale.

2. In this study, the UPGMA cluster analysis did not rely on gel banding from typing methods such as enterobacterial repetitive intergenic consensus (ERIC)-PCR, pulse field gel electrophoresis (PFGE), or restriction fragment length polymorphism (RFLP). We analyzed based on data from 1) Clermont phylogroup classification from PCR analysis that could identify each strain to be Clermont group (A, B, C, D, etc) and 2) UPEC and DEC virulence-associated gene profiles. These profiles were converted to binary data (presence <1> or absence <0>) for the UPGMA cluster analysis. Therefore, tolerance and optimization rate analysis were not applicable in this study. U20 and U19, U21 and U22 exhibited 100% similarity in virulence-associated gene profiling, suggesting they should be clustered together. However, they belonged to different phylogroups, resulting in their placement into separate groups. For example, U20 belonged to phylogroup C while U19 belonged to phylogroup B1. Consequently, these strains were assigned to different clusters despite sharing identical virulence-associated gene profiles due to their distinct phylogroups. Therefore, re-analysis of the dendrogram may not be necessary. 

I hope you understand our response.

Reviewer #6: 

Comment: At first, the authors did not adequately answer some questions. For example, one of the questions was why the researchers used the method of UPGMA clustering for cluster analysis and why the other clustering methods were not suitable. They suggested that due to the limited facilities, it was not possible to perform other tests such as a complete genome examination, which was a wrong answer to this question. In fact, they should have explained other clustering methods, and why they did not use those methods. Another example is my question “Have only the specific genes of UPEC been examined?” they didn’t answer it correctly. They stated that the purpose of their study was to examine virulence genes related to UPEC isolates, but they have other types of E. coli isolates. In this section, writing should have been done in such a way that the importance of all isolate types was also shown.

Responses: Thank you for your valuable comments. Please fine our responses below.

1. The reason to use the UPGMA clustering method for cluster analysis is because our data came from phylogrouping and virulence-associated genes determining which were binary data [absence (0) or presence (1)]. For binary data, hierarchical clustering methods like UPGMA and hierarchical agglomerative clustering (HAC) are commonly used to construct dendrograms. These two methods are suitable for binary data because they can utilize appropriate similarity index, such as the Jaccard index or Dice coefficient, to quantify the similarity between samples based on their binary profiles. UPGMA treats binary data more naturally by considering the presence or absence of traits. It computes distances between samples without evolutionary assumptions. In contrast, HAC may require additional preprocessing or transformation of binary data to compute meaningful distances between samples. Moreover, UPGMA produces dendrograms with a uniform branch length, which can be easier to interpret visually while HAC dendrograms may exhibit varying branch lengths depending on the chosen linkage criterion, offering more detailed insights into the clustering structure. In addition, UPGMA is relatively simple to implement and computationally efficient, especially for moderate-sized data sets compared to HAC that suitable for small to medium-sized data sets. Therefore, the UPGMA was suitable for clustering in this study.

2. In this study, we also determined the presence of virulence-associated genes associated with diarrheagenic E. coli in all 330 E. coli strains. Additional details regarding this aspect have been included in the abstract to provide further clarity on the scope of this study (Lines 26-30).

---

## [Decision Letter · Decision Letter 2]

9 Jul 2024

Phylogenetic diversity and virulence gene characteristics of Escherichia coli from pork and patients with urinary tract infections in Thailand

PONE-D-23-44006R2

Dear Dr. Suzuki,

We’re pleased to inform you that your manuscript has been judged scientifically suitable for publication and will be formally accepted for publication once it meets all outstanding technical requirements.

Kind regards,

Kwame Kumi Asare, Ph.D

Academic Editor

PLOS ONE

Additional Editor Comments (optional):

Reviewers' comments:

Reviewer's Responses to Questions

**Comments to the Author**

1. If the authors have adequately addressed your comments raised in a previous round of review and you feel that this manuscript is now acceptable for publication, you may indicate that here to bypass the “Comments to the Author” section, enter your conflict of interest statement in the “Confidential to Editor” section, and submit your "Accept" recommendation.

Reviewer #2: All comments have been addressed

Reviewer #4: All comments have been addressed

2. Is the manuscript technically sound, and do the data support the conclusions?

Reviewer #2: Yes

Reviewer #4: Yes

3. Has the statistical analysis been performed appropriately and rigorously? 

Reviewer #2: Yes

Reviewer #4: Yes

4. Have the authors made all data underlying the findings in their manuscript fully available?

Reviewer #2: Yes

Reviewer #4: Yes

5. Is the manuscript presented in an intelligible fashion and written in standard English?

Reviewer #2: Yes

Reviewer #4: Yes

6. Review Comments to the Author

Reviewer #2: (No Response)

Reviewer #4: The authors have responded appropriately to comments and incorporated feedback into the revised, and improved version of the manuscript.

7. PLOS authors have the option to publish the peer review history of their article (what does this mean?). If published, this will include your full peer review and any attached files.

Reviewer #2: No

Reviewer #4: **Yes: **Hami Kaboosi

---

## [Editor Report · Acceptance letter]

17 Jul 2024

PONE-D-23-44006R2 

PLOS ONE

Dear Dr. Suzuki, 

I'm pleased to inform you that your manuscript has been deemed suitable for publication in PLOS ONE. Congratulations! Your manuscript is now being handed over to our production team.

Kind regards, 

on behalf of

Dr. Kwame Kumi Asare 

Academic Editor

PLOS ONE